# LLC: Accurate, Multi-purpose Learnt Low-dimensional Binary Codes

**Aditya Kusupati[†],**
**Matthew Wallingford[†], Vivek Ramanujan[†], Raghav Somani[†], Jae Sung Park[†], Krishna Pillutla[†],**
**Prateek Jain[‡], Sham Kakade[†] and Ali Farhadi[†]**
[†]University of Washington, [‡]Google Research India
{kusupati,mcw244,ramanv,raghavs,jspark96,pillutla,sham,ali}@cs.washington.edu,
prajain@google.com

## Abstract

Learning binary representations of instances and classes is a classical problem with several high potential applications. In modern settings, the compression of high-dimensional neural representations to low-dimensional binary codes is a challenging task and often require large bit-codes to be accurate. In this work, we propose a novel method for **L**earning **L**ow-dimensional binary **C**odes (LLC) for instances as well as classes. Our method does *not* require any side-information, like annotated attributes or label meta-data, and learns extremely low-dimensional binary codes ($\approx 20$ bits for ImageNet-1K). The learnt codes are super-efficient while still ensuring *nearly optimal* classification accuracy for ResNet50 on ImageNet-1K. We demonstrate that the learnt codes capture intrinsically important features in the data, by discovering an intuitive taxonomy over classes. We further quantitatively measure the quality of our codes by applying it to the efficient image retrieval as well as out-of-distribution (OOD) detection problems. For ImageNet-100 retrieval problem, our learnt binary codes outperform 16 bit HashNet using only 10 bits and also are as accurate as 10 dimensional real representations. Finally, our learnt binary codes can perform OOD detection, out-of-the-box, as accurately as a baseline that needs $\approx 3000$ samples to tune its threshold, while we require *none*. Code is open-sourced at https://github.com/RAIVNLab/LLC.

## 1 Introduction

Embedding data in low-dimensional binary space is a long-standing machine learning problem [56]. The problem has received a lot of interest in the computer vision (CV) domain, where the goal is to find binary codes that capture the key semantics of the image, like, objects present in the image or interpretable attributes. Section 2 covers the literature on learning binary codes and their applications.

In addition to learning semantically meaningful representations of the instances, low-dimensional binary codes allow efficiency in a variety of large-scale machine learning (ML) applications. Low-dimensional codes are crucial in extreme classification with millions of classes [8, 30, 15] and also critical in efficient large-scale retrieval settings [38, 16, 59].

Compressing information into binary codes is challenging due to its highly non-smooth nature while requiring the preservation of relevant information in an instance/class. This might explain the lack of good classification accuracy for existing classification-based embedding techniques [28, 13]. To address that, traditional methods often relied on side-information like attributes to construct class codes and then use that to learn the instance codes [18, 1].

35th Conference on Neural Information Processing Systems (NeurIPS 2021).

Learning binary embeddings can be posed in a variety of formulations like pairwise optimization [34] or unsupervised learning [11, 50], in this work we focus on learning binary codes using a given labeled multi-class dataset, e.g., ImageNet-1K. This allows us to couple the representation (code) learning of both *instances* and *classes* thus enabling us to capture the underlying semantic structure efficiently to assist in downstream tasks like classification, retrieval etc.

We propose LLC, a method to learn *both* class and instance codes via the standard classification task and its setup *without any side-information*. Our Learning Low-dimensional binary Codes (LLC) technique, formulates the embedding (code) learning problem as that of learning a low-dimensional binary embedding of a standard deep neural "backbone". Instead of directly training for the low-dimensional binary instance codes, we propose a two-phase approach. In the first phase, LLC learns low-dimensional ($k$-bit) binary codes for classes that capture semantic information through a surrogate classification task. Then in the second phase, LLC uses these learnt class codes as an efficient alternative to learning instance codes in sub-linear cost (in the number of classes, $L$) using the Error-Correcting Output Codes (ECOC) approach [19]. This two-phase pipeline helps in the effective distillation of required semantic similarity between instances through the learnt class codes. For example, on ImageNet-1K with ResNet50, LLC is able to learn tight 20-bit codes that can be used for *efficient classification* and achieve 74.5% accuracy compared to the standard baseline 77% on ImageNet-1K (Section 4.1). Furthermore, we observe that the learnt 20-bit class codes capture intuitive taxonomy over classes (Figure 1) while the instance codes retain the distilled class similarity information useful in efficient retrieval and OOD detection.

**Retrieval.** To further study, the effectiveness of our learnt binary codes, we apply them to hashing-based efficient retrieval, where the goal is to retrieve a large number of similar instances with the same class label in top retrieved samples. Deep supervised hashing is a widely studied problem with several recent results [9, 53] which are designed *specifically* for the learnt hashing-based retrieval. Interestingly, our learnt instance codes through the LLC routine provide strikingly better performance while not being learnt explicitly for hashing. For eg., using AlexNet, with just 32-bit codes we are can provide 5.4% more accurate retrieval than HashNet's 64-bit codes on ImageNet-100 (Section 4.2).

**OOD Detection.** We similarly apply LLC based learnt binary codes to detect OOD instances [26]. We adopt a simple approach based on our binary codes: if an instance is not within a Hamming distance of 1 to any class codes, we classify it as OOD. That is, we do not fine-tune our OOD detector for the new domain, which is critical in practical settings. In contrast, baseline techniques for OOD detection require a few samples (eg., $\approx 3000$ for ImageNet-750) from the OOD domain to fine-tune thresholds, while we require *no* samples yet reaching similar OOD detection (Section 4.3).

In this work, we make the following key contributions:

- LLC method to learn semantically-rich low-dimensional binary codes for both classes & instances.
- Show that the learnt codes enable accurate & efficient classification: ImageNet-1K with 20-bits.
- Apply LLC to image retrieval task, and demonstrate that it comfortably outperforms the instance code learning methods for hashing-based retrieval on ImageNet-100.
- Finally, use codes from LLC for strong & sample efficient OOD detection in practical settings.

## 2 Related Work

Binary class codes were originally aimed at sub-linear training and prediction for multi-class classification. The Error-Correcting Output Codes (ECOC) framework [19, 3, 20] reformulated multi-class classification as multi-label classification using $k$-bit codes per class (codebook). The learning of optimal codebook is NP-complete [14] which lead to use of random codebooks [28, 13] in traditional ML. However, there were a few codebook learning [5, 64, 58, 4] and construction schemes using side-information from other modalities [1]. The lack of a strong learnable feature extractor often deterred the gains these codebooks provide for the classification and effective learning of instance codes. Attribute annotations can also help in constructing class codes [2]. These binary codes are either explicitly annotated [21] or discovered [47, 22]. Attributed-based learning also ties into leveraging the class codes for zero/few-shot learning [36, 37, 1, 44] expecting some form of interpretability.

Most methods that use class codes as supervision can produce instance codes [19]. However, the standalone literature of instance codes comes from requirements in large-scale application like retrieval (hashing). In the past, most hashing techniques that created instance codes were based on

random projections [16, 13, 12], semantics [51, 18] or learnt through metric learning [34, 33, 43, 32], clustering [57, 50] and quantization [23]. Deep learning further helped in learning more accurate hashing functions to generate instance codes either in an unsupervised [11, 52] or supervised [38, 9, 53, 62] fashion. We refer to [39, 63, 56] for a more thorough review on deep hashing methods.

Finally, embedding-based classification [13, 61, 8, 24] enables joint low-dimensional representation learning for both classes and instances with an eye on sub-linear training and prediction costs. After distilling the key ideas from the literature, we aim to a) learn semantically rich low-dimensional representations for both classes and instances together, b) have these representations in the binary space, and c) do this with minimal dependence on side-information or metadata.

LLC, to the best of our knowledge - for the first time, jointly learns low-dimensional binary codes for both classes and instances using a surrogate classification task, without any side-information (Section 3). The learnt class codes capture intrinsic information at the semantic level that helps in discovering an intuitive taxonomy over classes (Figure 1). The learnt class codes then anchor the instance code learning which results in tight and accurate low-dimensional instance codes further used in retrieval (Section 4.2). Finally, both the learnt class and instance codes power extremely efficient yet accurate classification (Section 4.1) and out-of-distribution detection (Section 4.3).

## 3 Learning Low-dimensional Binary Codes

The goal is to learn a binary embedding (code) function $g\colon \mathcal{X} \to \{-1, 1\}^k$ where $\mathcal{X}$ is the input domain and $k$ is the dimensionality of the code. We focus on learning embeddings using a labelled multi-class data [28]. That is, suppose we are given a labelled dataset $\mathcal{D} = \{(x_1, y_1), \ldots, (x_n, y_n)\}$ where $x_i \in \mathcal{X}$ is an input point and $y_i \in [L]$ is the label of $x_i$ for all $i \in [n]$. Then, the goal is to learn an instance embedding function $g\colon \mathcal{X} \to \{+1, -1\}^k$ *and* class embeddings $h_q \in \{+1, -1\}^k$ for all $q \in [L]$ such that $g(x_i) = h_{y_i}$ and $g(x_i) = g(x_j)$ if and only if $y_i = y_j$.

Intuitively, for large-scale datasets, $g(x)$ and $h_q$ should capture key semantic information to provide accurate classification, thus allowing their use in application domains like retrieval or OOD detection. Note that while we present our technique for learning embeddings using multi-class datasets, it applies more generally to multi-labeled datasets as well.

**Instance and Class Code Parameterization.**  For learning such embedding function, we assume access to a deep neural architecture $F(\,\cdot\,; \theta_F)\colon \mathcal{X} \to \mathbb{R}^d$ that maps the input $x \in \mathcal{X}$ to a $d$-dimensional real-valued representation. $\theta_F$ is a learnable parameterization of the network; we drop $\theta_F$ from $F$ wherever the meaning is clear from the context. For example, ResNet50 is one such network that encodes $224 \times 224$ RGB images into $d = 2048$ dimensions.

Now, given a network $F$ and $x \in \mathcal{X}$, we formulate embedding function of $x$ and the corresponding multiclass prediction scores $\hat{y} \in \mathbb{Z}^L$ as:

$$g(x) \coloneqq B\left(\mathbf{P} \cdot F(x; \theta_F)\right), \quad \hat{y} \coloneqq B(\mathbf{C}) \cdot g(x), \tag{1}$$

where $\mathbf{P} \in \mathbb{R}^{k \times d}$ maps $F(x)$ into $k$-dimensions and $B(a) = \mathrm{sign}(a) \in \{+1, -1\}$ is the standard binarization/sign operator applied elementwise (with the assumption $\mathrm{sign}(0) = +1$). Finally, $\mathbf{C} \in \mathbb{R}^{L \times k}$, and $\hat{y} = B(\mathbf{C}) \cdot g(x)$ represents the scores of each class for an input $x$. Note that for a class $\ell \in [L]$, $B(\mathbf{C}_\ell)$ (where $\mathbf{C}_\ell$ represents the $\ell$-th row of $\mathbf{C}$) is the learnt binary class embedding (code) of class $\ell \in [L]$, and $g(x) = B(\mathbf{P} \cdot F(x; \theta_F))$ is the learnt instance embedding (code) of instance $x$. Note that (1) is a general purpose formulation for the problem of learning class and instance codes.

### 3.1 The LLC Method

**Phase 1: *Codebook Learning* – $B(\mathbf{C})$.**  Given labelled examples $\mathcal{D}$, we use standard empirical risk minimization to learn a multi-class classifier, i.e., we solve

$$\min_{\mathbf{C}, \mathbf{P}, \theta_F} \sum_{(x_i, y_i) \in \mathcal{D}} \mathcal{L}\left(B(\mathbf{C}) \cdot (\mathbf{P} \cdot F(x_i; \theta_F)) ; y_i\right), \tag{2}$$

where $\mathcal{L}\colon \mathbb{R}^L \times [L] \to \mathbb{R}_+$ is the standard multi-class softmax cross-entropy loss function. This is a standard optimization problem that can be solved using standard gradient descent methods or other

sub-gradient based optimizers. However, one challenge is that $B(\mathbf{C})$ is a binary matrix and $B$ is a binary function, so the gradients are 0 almost everywhere. Instead, we use the Straight-Through Estimator (STE) [6] technique popular in binary neural networks domain [48], to optimize for $\mathbf{C}$ through the binarization. Intuitively, STE uses binarization/sign function in the forward pass, but in the backpropagation phase, it allows the gradients to flow straight-through as if it were real-valued. The codebook, $B(\mathbf{C})$ refers to the collection of $k$-bit class codes learnt in this process.

For ImageNet-1K, we learnt unique binary codes, $B(\mathbf{C}_\ell)$, for every class $\ell \in [L]$ of the 1000 classes using only 20-bits, only twice the information-theoretic limit. As with the class representations from a linear classifier, these class codes do capture intrinsically important features that help in discovering intuitive taxonomy over classes (Section 3.2) among various applications (Section 4).

**Phase 2: *Instance Code Learning* –** $B(\mathbf{P} \cdot F(x; \theta_F))$.    Several existing techniques model $\mathbf{C}$ and $\mathbf{P}$ in different ways to learn an embedding function similar to (1). However, these methods often try to only learn instance codes and have challenges in maintaining high accuracy [9, 11] in a variety of applications because optimization problem (2) is challenging and might lead to significantly sub-optimal classification error. For example, for ImageNet-1K classification with ResNet50, the accuracy for our trained model (20-bits) at this stage is 72.5% compared to the standard 77%.

To remedy this, we further optimize our embeddings using the ECOC framework [19] for multi-class classification, which essentially transforms the multi-class problem into a multi-label problem, which in turn is $k$ independent binary classification problems. That is, we use the $k$-bit codes learnt for each class as the supervision to further train $F(\,\cdot\,; \theta_F)$ and $\mathbf{P}$:

$$\min_{\theta_F, P} \sum_{(x_i, y_i) \in \mathcal{D}} \sum_{j=1}^{k} \mathrm{BCE}\left(\sigma(\mathbf{P}_j \cdot F(x_i; \theta_F))\,;\, (B(\mathbf{C}_{y_i, j}) + 1)/2\right), \qquad (3)$$

where $\sigma$ is the sigmoid/logistic function, BCE is the binary cross-entropy loss between the $j$-th bit of instance $x_i$'s embedding, and the $j$-th bit extracted from the class embedding of it's label $y_i$ (the function $z \mapsto (z+1)/2$ is used to map $\{+1, -1\}$ to $\{1, 0\}$ to make it a simple binary classification problem per each bit). We use gradient based optimization to learn $\theta_F$ and $\mathbf{P}$. As mentioned earlier, ECOC framework allows us to correct errors in classification. For example, with just 20 bits on ImageNet-1K dataset, the method now achieves 74.5% accuracy with ResNet50 backbone.

The advantage of this two-phase pipeline where we start with a codebook learning for classes is that the cost of learning instances codes reduces to a bottleneck of $k$-dims ($\ll L$) instead of the usual $L$ . Furthermore, these learnt low-dimensional binary codes for both classes and instances help in large-scale applications via efficient classification and retrieval (see Section 4). Note that, unlike attribute-based methods [37], we do *not* require additional meta-data, but learn binary codes by only using the standard classification task. This also circumvents the potential instabilities of pairwise optimization in instance binary code learning which often leads to poor class codes due to codebook collapse. At the end of LLC routine, we have learnt the instance codes, $B(\mathbf{P} \cdot F(x; \theta_F))$, and class codes, $B(\mathbf{C})$ to be used for downstream applications. Algorithm 1 presents LLC in full.

---

**Algorithm 1** The LLC Method

---

**Input:** $\mathcal{D}$, $F$ and $B$
**Output:** $\mathbf{C}$, $\mathbf{P}$ and $\theta_F$
 1: **Codebook Learning –** $B(\mathbf{C})$: Solve (2) using ERM and STE to get $\mathbf{C}$, $\mathbf{P}$ and $\theta_F$ -

$$\mathbf{C}, \mathbf{P}, \theta_F \leftarrow \arg\min_{\mathbf{C}, \mathbf{P}, \theta_F} \sum_{(x_i, y_i) \in \mathcal{D}} \mathcal{L}\left(B(\mathbf{C}) \cdot (\mathbf{P} \cdot F(x_i; \theta_F))\,;\, y_i\right)\ .$$

 2: **Instance Code Learning –** $B(\mathbf{P} \cdot F(x; \theta_f))$: Further optimize $\mathbf{P}$ and $\theta_F$ by solving (3) using ECOC framework and ERM by fixing $\mathbf{C}$ -

$$\theta_F, P \leftarrow \arg\min_{\theta_F, P} \sum_{(x_i, y_i) \in \mathcal{D}} \sum_{j=1}^{k} \mathrm{BCE}\left(\sigma(\mathbf{P}_j \cdot F(x_i; \theta_F))\,;\, (B(\mathbf{C}_{y_i, j}) + 1)/2\right)\ .$$

---

Overall, we present a simple yet scalable method to learn low-dimensional (exact) binary codes for both classes and instances which in turn could power multi-class classification with sub-linear costs (in terms of $L$) and efficient retrieval for large-scale applications. Using our method, we can consistently learn unique low-dimensional binary codes for all 1000 classes in ImageNet-1K using only 20-bits (which is twice the information-theoretic limit of $\lceil \log_2(1000) \rceil$). Next, we discuss the learnt codebook's intrinsic information about the classes and their structure.

## 3.2 Discovered Taxonomy and Visualizations

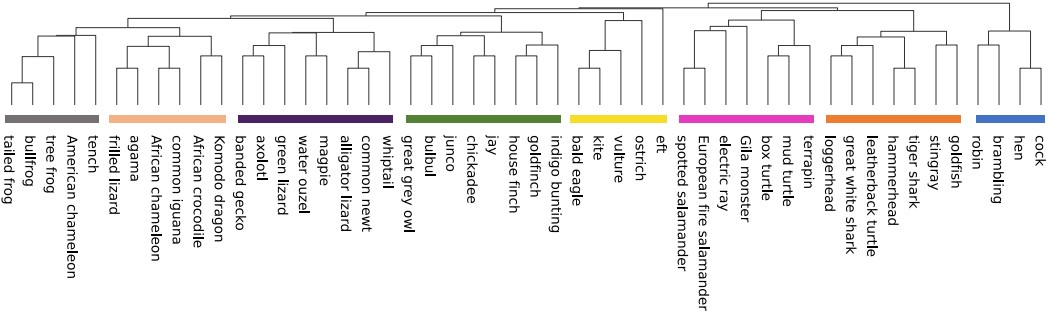

Figure 1: Discovered taxonomy over 50 classes of ImageNet-1K using the learnt 20-bit class codes. Related species are well clustered while pushing away unrelated ones. Figure 3 in Appendix D contains the codebook.

After learning the 20-bit binary codebook for 1000 classes of ImageNet-1K, we used the class representation from $B(\mathbf{C})$ of the first 50 classes to discover an intuitive taxonomy through agglomerative clustering [42]. Figure 1 shows the discovered hierarchy. This hierarchy effectively separates birds from amphibians; frogs and chickens are on extremes of the taxonomy and brings species with shared similarities closer (lizards & crocodiles; marine life). While the taxonomy is not perfect, the 20-bits do capture enough important information that can be used downstream.

Figure 2 shows the pair-wise inner-product heat maps for all the 1000 classes using 20-bits and 2048-dimensional real representation; the comparison reveals that 20-bits indeed highlights the same substructures as the higher dimensional real-valued embeddings. Appendix D has a more detailed discussion about quantitatively evaluating the discovered hierarchy and more visualizations.

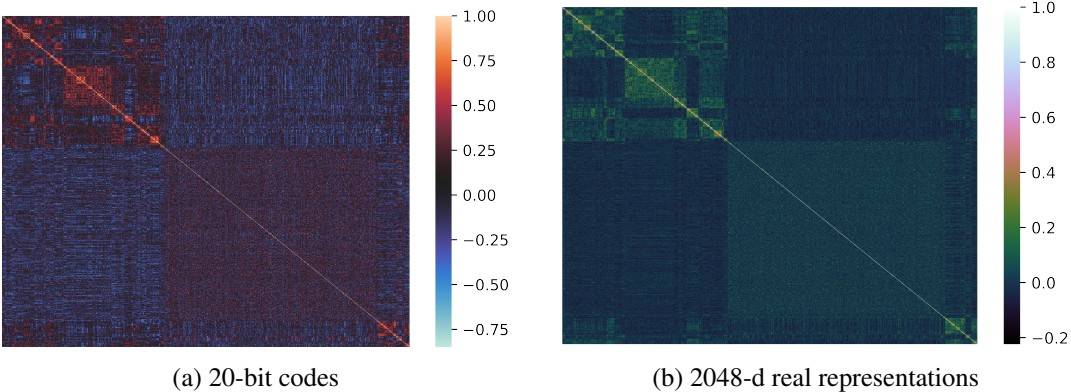

(a) 20-bit codes  (b) 2048-d real representations

Figure 2: The pair-wise inner product heat maps of class representations a) learnt 20-bit codes & b) learnt 2048 dimensional real representations for the 1000 classes in ImageNet-1K. Similar sub structures are highlighted in both heatmaps and often correspond to local hierarchy present in the classes thus making a case that 20-bit codes distill enough information to capture hierarchy of the classes.

## 4 Applications

In this section, we discuss three applications of the learnt low-dimensional binary codes: 1) efficient multi-class classification (Section 4.1), 2) efficient retrieval (Section 4.2), and 3) out-of-the-box

out-of-distribution (OOD) detection (Section 4.3). We also present ablation studies on codebook learning, feature separability and classification (Section 4.4).

## 4.1   Efficient Multi-class Classification

Recall that the proposed LLC algorithm outputs a) the learnt class codes (codebook), $B(\mathbf{C})$ and b) an *encoder* that produces instance codes, $B(\mathbf{P} \cdot F(\mathbf{x}; \theta_F))$ for $x$. We define a class codebook as a collection of $L$ binary vectors, one for each class in the dataset, that can then be used for classification of a test instance $x$. We can use several "decoding" routines to classify an instance $x$, given its encoding and the learnt codebook. Below we discuss two decoding schemes that are diametrically opposite in terms of the inference cost. Also, note that the standard linear classification with real-valued representation and classifiers scale as $O(L)$ in terms of computational complexity and model size.

### 4.1.1   Decoding Schemes

**Exact Decoding (ED).**   Exact Decoding scheme expects the Hamming distance between the generated instance code, $B(\mathbf{P}F(\mathbf{x}; \theta_F))$, and the ground truth class code, $B(\mathbf{C}_i)$ to be *exactly* 0. That is, we can hash the class codes in a table, and then ED requires only a $O(1)$ hash-table lookup for a given instance. Consequently, the inference time for ED is nearly *independent* of $L$. Naturally, the decoding scheme is highly stringent and would misclassify an instance if the instance binary code and the ground truth code do not match in even a single *bit*. Surprisingly, this highly efficient decoding scheme still provides non-trivial accuracy (see Table 1 and Section 4.1.2).

**Minimum Hamming Decoding (MHD).**   Minimum Hamming Decoding is akin to the Maximum Dot Product used by standard linear classifiers. For an instance code, we evaluate the Hamming distance with all the $L$ class codes and output the class with the least Hamming distance. Note that the Hamming distance over binary codes can be computed using XOR operations that are implemented significantly more efficiently than the floating-point operations [48]. Even though, technically, computational complexity and model size of MHD scales as $O(L)$ but the real-world implementations should be an order of magnitude faster than standard classifiers. In fact, for large number of classes $L$, the efficiency of MHD can be further improved by using approximate nearest neighbour search [16, 7, 40]. Appendix A has the mathematical presentation of the decoding schemes.

See Section 5 for more discussion on potential decoding schemes. Also see Section 4.4 for ablation studies about the two decoding schemes along with feature separability (linear vs Hamming).

### 4.1.2   Empirical Evaluation

ImageNet-1K [49] is a widely used image classification dataset with 1000 hierarchical classes. Our classification experiments use ResNet50 [25] and are trained using the $\sim$1.3M training images. Images were transformed & augmented with standard procedures [35, 60]. All the implementations were in PyTorch [45] and experimented on a machine with 4 NVIDIA Titan X (Pascal) GPUs.

When applied to ImageNet-1K, the first phase of LLC, learnt a 20-bit codebook with 1000 unique class codes, i.e., every class has its own *distinct* binary code. We warm start the second phase of LLC by the learnt ResNet50 backbone along with the 20 dimensional projection layer. See Appendix C for the hyperparameter values and other training details.

A key feature of LLC is that it jointly learns both the class codebook as well as instance codes. Several existing techniques decouple this learning process where the codebook is constructed separately and is then used to train the instance codes [28, 14, 3, 20, 1, 58]. We evaluate the advantage of the *joint* learning approach of LLC by comparing its performance against three strong baselines: i) Random codebook of 20-bits, ii) 20-bit CCA codebook [1, 58, 64] & iii) 20-bit SVD codebook. Previous works [28, 14, 3] argued that random codebooks are competitive to the ones constructed using side-information. 20-bit CCA and SVD codebooks aim to capture the hierarchy that is amiss in the random codebook. The 20-bit SVD codebook is built using the SVD of 2048 dimensional linear classifiers (for each class) in the pre-trained ResNet50, and binarizing it. 20-bit CCA codebook is the binarized version of the transformed label embedding projected on to 20 components learnt using CCA between 2048 dimensional representations of 50K samples from the ImageNet train set and

Table 1: Classification performance on ImageNet-1K with ResNet50 using various class codebooks for training.

| Codebook | Unique Codes | ED Accuracy (%) | MHD Accuracy (%) |
|---|---|---|---|
| Random 20-bits | 1000 | 64.07 | 66.91 |
| CCA 20-bits | 813 | 55.17 | 57.03 |
| SVD 20-bits | 969 | 65.12 | 69.18 |
| LLC 20-bits (Ours) | 1000 | **68.82** | **74.57** |

Table 2: Classification accuracy on ImageNet-1K vs. bit length of the learnt class codebooks (§4.4).

| LLC Length | Unique Codes | ED Accuracy (%) | MHD Accuracy (%) |
|---|---|---|---|
| 15 bits | 990 | 67.20 | 71.03 |
| 20 bits | 1000 | **68.82** | 74.57 |
| 25 bits | 1000 | 67.93 | 74.79 |
| 30 bits | 1000 | 67.51 | **75.13** |

their one-hot label embeddings. Despite being able to capture the hierarchy information, both 20-bit CCA/SVD codebooks suffer from clashes reducing their overall effectiveness.

Next, using the baselines codebooks and the corresponding learnt instance codes, we compute class predictions for each test instance using the Exact Decoding (ED) & Minimum Hamming Decoding (MHD) schemes mentioned in the previous section. We evaluate all the methods using top-1 accuracy on the ImageNet-1K validation set. Baseline ResNet50 architecture represents the maximum accuracy we can hope to achieve using binarized instance+class codes. Note that this baseline classifier requires $O(L)$ computation over 16-bit real numbers, and achieves Top-1 accuracy of 77%.

Table 1 compares the accuracy of LLC (with 20-bit codebook) against baseline codebooks mentioned above. Note that MHD with LLC codebook is 74.5% accurate, i.e., despite using only 20-dimensional *binary* representation it is only about 2.5% less accurate than standard ResNet50 that uses 2048 dimensional real-valued representation. Furthermore, we observe that compared to standard codebooks like SVD, our jointly learnt codebook is 5% more accurate.

Interestingly, Exact Decoding (ED) – which is $O(1)$ inference scheme – with LLC codebook is nearly as accurate as the SVD codebook with MHD scheme and is about 12% more accurate than the CCA codebook with ED scheme. Naturally, codebook length/dimensionality plays a critical role in classification accuracy; see Section 4.4 for a detailed ablative study on this aspect. Finally, the gains in efficiency should be even more compelling for problems with millions of classes [54].

## 4.2 Efficient Retrieval

The goal in retrieval is to find instances from a database that are most similar to a given *query*. Traditional retrieval approaches, use a *fixed metric* to retrieve "similar points", with data structures like LSH for efficient retrieval. Recent progress in Deep Supervised Hashing (DSH) [38] offer significantly more compelling solutions by learning the hashing function itself. That is, DSH aims to learn binary codes for each instance s.t. a pair of instances are embedded closely iff they belong to the same class, and then learns the hashing function end-to-end using a small train set.

As LLC also learns instance codes to reflect class membership, we can directly use our learnt encoder as a hashing function for given instances. For each query, the most relevant samples from the database are retrieved based on the minimum Hamming distance. Similar to the decoding schemes in classification, the retrieval can be optimized using approximate nearest neighbor search. Finally, the efficiency gains provided by using bits instead of real numbers should enable deployment of LLC based retrieval for efficient high recall portions of retrieval pipelines.

### 4.2.1 Empirical Evaluation

Following DSH literature, we evaluate hashing-based image retrieval on ImageNet-100, a benchmark dataset created by Cao et al. [9]. ImageNet-100 has 100 classes randomly sampled from ImageNet-1K. All the validation images of these classes are used as query images, all the training images ($\sim 1300$ per class) of these 100 classes are used as database images. Finally, 130 samples per class from the database are used as the training set for learning binary codes or hashing functions.

We compare against HashNet [9] and Greedy Hash [53] for image retrieval using learnt instance codes. HashNet learns the bit representations of instances using a pairwise optimization with positive and negative instance pairs. HashNet is a representative baseline for an alternative way of learning binary instance codes compared to LLC. On the other hand, Greedy Hash learns only the instance codes using straight-through-estimator via the classification task. Note that LLC learns both class

Table 3: Efficient image retrieval on ImageNet-100 using AlexNet compared using MAP@1000 (Appendix B) across 16 – 64 bits.

| Method | 10 bits | 16 bits | 32 bits | 48 bits | 64 bits |
|---|---|---|---|---|---|
| HashNet [9] | 0.1995 | 0.2815 | 0.4300 | 0.5270 | 0.5124 |
| Greedy Hash [53] | 0.2860 | 0.4247 | 0.5412 | 0.5720 | 0.5895 |
| LLC (Ours) | **0.3086** | **0.4305** | **0.5565** | **0.5749** | **0.6000** |

Table 4: Comparison of LLC based retrieval vs real-valued representations with ResNet50 on ImageNet-100 using MAP@1000.

| Representation | 8 dims | 10 dims | 64 dims |
|---|---|---|---|
| LLC (1 bit) | - | 0.6458 | 0.6773 |
| Real (16 bits) | 0.5041 | 0.6657 | 0.7794 |

codes as well as instance codes differentiating it from Greedy Hash style methods. Learnt instance codes are a byproduct of efficient classification as opposed to baselines that optimize for them.

We use the Mean Average Precision (MAP@1000) metric for evaluation. The MAP@1000 calculation code of HashNet [9] is erroneous and has propagated to several papers in the literature. We use the corrected version, hence the accuracy numbers are different from the original paper. Please see Appendix B for the corrected version, the changes required along with an example and a brief discussion. We used the publicly available pre-trained HashNet models [10] and Greedy Hash [53] code to recompute the MAP@1000.

Following HashNet [9], we use AlexNet [31] as the backbone and warm-start it with a pre-trained model on ImageNet-1K. We add a projection layer to the backbone and learn the instance and class codes. We also report retrieval numbers with ResNet50 [25] and compare LLC based retrieval numbers to learnt real-valued representations. Please see Appendix C for the training details and hyperparameters of efficient retrieval pipelines.

Table 3 shows the performance (evaluated using MAP@1000) for HashNet, Greedy Hash, and LLC across various code lengths. LLC outperforms HashNet across all code lengths (16 – 64) by at least 4.79% on MAP@1000. LLC is also better than Greedy Hash across all the bit lengths. LLC also outperforms 16-bit HashNet by 2% & 15% using only 10 & 16 bits respectively. Finally, 32-bit LLC comfortably outperforms both 48 & 64-bit HashNet showcasing the effectiveness of our learnt tight bit codes. Note that LLC, learning both instance and class codes, is effective in retrieval even though it was designed for classification.

We repeat the retrieval experiments with ResNet50. Table 4 shows the MAP@1000 for LLC with 10 and 64 bits along with the same dimensional real-valued representations. The 10-bit LLC is only 2% lower than 10 dimensional real-valued representation even though theoretically, the cost associated with 10-bit LLC based retrieval is about $256\times$ less than 10 dimensional real representations.

The 64 bit and 10 bit LLC outperforms 10 and 8 dimensional real-valued representations respectively at a much cheaper retrieval cost, at least by an order of magnitude. More discussion about the use of binary codes for retrieval at a large scale can be found in Section 5. Finally, 10-bit LLC with ResNet50 outperforms the best performing AlexNet based models for the same task, suggesting ResNet50 is a more appropriate architecture for benchmarking DSH literature.

### 4.3 Out-of-Distribution (OOD) Detection

For a multi-class classifier, detecting an OOD sample is very important for robustness [26] and sequential learning [55]. Multi-class classifiers are augmented with OOD detection capability by setting a threshold on heuristics like maximum logit which is tuned using a validation set.

We focus on the scenario where the ratio of in-distribution to out-of-distribution samples in the validation set is not representative of the deployment. This throws off the methods that try to maximize metrics, F1, using a validation set. Our learnt class codebook from LLC comes with over-provisioning (for ease of optimization) resulting in unassigned codes. These unassigned codes can be treated as OOD out-of-the-box with no tuning whatsoever. That is, we classify an instance as OOD if its instance code does not match *exactly* with the code of a class in our learnt codebook.

Appendix E discusses the OOD detection experiments on ImageNet-750 [55] & MIT Places [65]. At a high level, LLC based out-of-the-box OOD detection (with a learnt 20-bit codebook on ImageNet-1K) achieves nearly the same OOD detection accuracy as a baseline [26] that tries to maximize F1 using a validation set. We would like to stress that while such a method needs $\approx 3000$ points in the validation set, our method requires *no* samples, which is critical in several practical settings.

## 4.4 Ablation Studies

**Classification Accuracy vs Number of Bits.** Table 2 shows the trade-off in classification accuracy with the variation in the length of the learnt codebook for ImageNet-1K. LLC learns a 15-bit codebook with only 990 unique codes leading to a loss of accuracy due to code collapse in both ED and MHD schemes (1.62% & 3.54% compared to 20-bit codebook respectively). An interesting observation is that the ED accuracy gradually goes down after 20-bits while the MHD accuracy keeps on increasing. The phenomenon of increasing accuracy with MHD is probably due to the increase in the capacity of both instance and class codes. However, the decrease in ED accuracy after 20-bits can be explained through the hardness in exactly predicting every bit in the instance code to match the ground truth class code. Our classification model with 20-bits on average gets 19.2 bits correct but the model with 30-bits only gets 28.5 bits right. This increase in uncertainty coupled with the stringent ED scheme leads to a slight dip in accuracy as the code length increases. However, this also provides us with a path for more accurate decoding schemes while being efficient as discussed in Section 5.

**Classification Accuracy vs Faster Codebook Learning.** Codebook learning phase of LLC is expensive, this motivated us to speed up codebook learning at a minimal loss in accuracy. One way is to warm-start the codebook using the ones built with SVD/CCA (see Section 4.1). While these codebooks suffer from code collapse, with further training, they start to learn 1000 unique codes quickly. Using these final codebooks gets to a comparable (1% drop) accuracy as the 20-bit learnt LLC codebook but at a relatively cheaper training. Another option is to use only a portion of the data and a much smaller network to learn the codebook. We sampled 50K training images and use a MobileNetV1 [27] (which has about $6\times$ less parameters and compute than ResNet50) to learn a 20-bit codebook which gets to ED and MHD accuracy of 66.62% & 72.55% which is only 2% lower than the end-to-end learnt codebook but at a fraction of the training cost (3 hrs vs 2 days).

**Linear vs Hamming Separability.** Fitting a deep neural network to the learnt codebook for classification results in warping of the feature space considerably. The final classification space is a hypercube with the vertices being apart by Hamming distance of 1. To verify linear separability, we take the learnt, frozen ResNet50 trained for the 20-bit classification problem and fit a linear classifier on top of the 2048 dimensional features. Linear classifier quickly reaches a top-1 accuracy of 75.51%.

The opposite does not seem to be true. We extract and freeze the backbone of a pre-trained ResNet50 and train a projection layer to fit the 20-bit learnt codebook. This gets to top-1 accuracy of only about 21% with the ED scheme. However, we also observed that unfreezing and finetuning the last 3 layers of the backbone recovers the top-1 ED accuracy to roughly 68%.

These experiments show 1) Hamming separability inherently enables linear separability, 2) Linear separability does not imply Hamming separability & 3) with enough overparameterization, linearly separable space can be warped to support Hamming separability. Hamming separability automatically provides linear separability with increased accuracy of $\sim 1\%$ over the MHD scheme which allows for an option for using a more powerful yet simple classifier, in case of accuracy requirements.

**Independent vs Nested Codebook Learning.** Consider a scenario with varied computational budgets for classification. We could either train independent $k$-bit models (eg., $k = 20, 25, 30$) and use them according to the budget, or we could learn a single nested codebook-based model that can be readily adapted to any of these settings. While training a codebook of larger bit length like $k = 30$, we can ensure that the first $m$-bits, $m < k$, also form a codebook at minimal additional cost. We were able to stably train a 30-bit codebook and also extract 20, 25-bit codebooks from it all of which are as accurate as independently trained codebooks. These nested codebooks have the potential to be used based on the computational resource availability for efficient classification without having to retrain.

## 5 Discussion and Conclusions

We designed LLC to learn low-dimensional binary codes for instances as well as classes, and utilized them in applications like efficient classification, retrieval, OOD detection. A key finding is that combining class code learning with ECOC framework to learn instance code leads to a stable training system that can accurately capture the semantics of the class data despite just 20-dimensional code. Traditionally, methods like HashNet, KLSH [33] attempt to learn hashing function using pairwise

loss functions by embedding instances such that points from the same class are embedded closely and points from different classes are far. But such formulations are hard to optimize, due to the risk of embedding collapse. We observe that LLC by using instance-wise formulation can train stably with significantly higher performance. Another fascinating observation is that while architectures like ResNet50 have large intermediate (2048 dimensional real) representations, they can be compressed to just 20 bits without significant loss in accuracy! Even though quantization [48] literature demonstrates strong compression of representations, we believe such stark compression has been elusive so far and is worth further exploration from the efficient inference viewpoint.

**Limitations.** Our visualization (see Figure 4 in Appendix D) indicates that each bit does not correspond to some easily interpretable attributes, unlike DBC [47]. We believe incorporating priors with weak supervision as well as cross-modal learning could help LLC get past this limitation.

ED and MHD schemes are on two ends of the computation vs accuracy spectrum and do not transition smoothly. Designing decoding schemes that can compromise between these two extreme decoding schemes might be able to address this limitation.

Strong encoders are needed to warp the feature space to ensure Hamming separability. For example, using a 20-bit learnt codebook with ED scheme, ResNet50 gets to $68.8\%$ top-1 accuracy whereas MobileNetV1 can only reach $53.23\%$. This also ties into the argument that classification is a trade-off between encoder and decoder costs. Making decoders efficient and cheap, puts the burden on encoding the information in the right way and higher expressivity often helps in that cause.

**Future Work.** There are several exciting directions that we would like to explore. In principle, LLC can easily incorporate side-information when needed with simple additional losses during training. The additional regularization losses can also help in incorporating natural constraints on the codebook [14, 3] or can enable attribute-based class codes for interpretablity [21, 22, 37] making them exciting directions to explore. LLC algorithm can also be used to encode instances of multiple modalities like audio, visual, language to the same learnt low-dimensional binary space. This might help in effective cross-modal supervision along with retrieval among various other applications.

**Potential in Large-Scale Applications.** While our focus was on designing low-dimensional accurate binary codes, several studies [48, 29] have shown that efficiency afforded by bit-wise computation over floating-point computation can lead to almost an order of magnitude speed-up. Furthermore, as the number of classes increases, the learning of class codebooks helps in training representations in sublinear costs [13] along with sublinear inference (in $L$). We expect LLC algorithm to have its efficiency benefits outweigh the accuracy drop for large multi-class/multi-label problems, like objection recognition using ImageNet-22K [17], document tagging [54, 46] and instance classification [59]. The efficiency aspect of the binary codes has not been fully explored in this paper as the main computational bottleneck for ImageNet-1K classification is the deep neural network featurizer.

Lastly, LLC based efficient retrieval can be used for the initial high-recall shortlisting of a search pipeline, which is followed by high precision models operating on more expressive yet expensive embeddings. We leave practical demonstration of such a system at web-scale for future work.

## Acknowledgments

We are grateful to Tapan Chugh, Kunal Dahiya, Max Horton, Sewoong Oh, Mohammad Rastegari, Ludwig Schmidt and members of RAIVN Lab for helpful discussions and feedback. AK would also like to thank Soumen Chakrabarti and Manik Varma for sowing the seeds of this idea in his initial research days. Sham Kakade acknowledges funding from NSF Awards CCF-1703574. Ali Farhadi acknowledges funding from the NSF awards IIS 1652052, IIS 1703166, DARPA N66001-19-2-4031, W911NF-15-1-0543 and gifts from Allen Institute for Artificial Intelligence.

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
