# A    LLC **Decoding Schemes for Classification & Binarization Function**

---

**Algorithm 2** Inference using Exact Decoding (ED)

---

**Input:** $x \in \mathcal{X}$, $F(\cdot\,; \theta_F)$, $\mathbf{C}$ and $\mathbf{P}$
**Output:** $\ell^* \subseteq [L]$
  1: $g(x) \leftarrow B(\mathbf{P} \cdot F(x; \theta_F))$
  2: $\ell^* \leftarrow \{\ell \in [L] : B(\mathbf{C}_\ell) = g(x)\}$

---

---

**Algorithm 3** Inference using Minimum Hamming Decoding (MHD)

---

**Input:** $x \in \mathcal{X}$, $F(\cdot\,; \theta_F)$, $\mathbf{C}$ and $\mathbf{P}$
**Output:** $\ell^* \subseteq [L]$ ($\ell^* \neq \emptyset$)
  1: $g(x) \leftarrow B(\mathbf{P} \cdot F(x; \theta_F))$
  2: $\ell^* \leftarrow \arg\min_{\ell \in [L]} \frac{1}{2} \| B(\mathbf{C}_\ell) - g(x) \|_1$

---

---

**Algorithm 4** PyTorch code for Binarization $B(\cdot)$ function with straight-through-estimator (STE).

---

```python
class Binarize(autograd.Function):
    @staticmethod
    def forward(ctx, weight):
        out = weight.clone()
        # binarizing in the forward pass
        out[out >= 0] = 1
        out[out < 0] = -1

        return out

    @staticmethod
    def backward(ctx, g):
        # send the gradient g straight-through on the backward pass.
        return g, None
```

---

# B    Corrected MAP@1000 Metric for Image Retrieval.

The reported MAP@1000 metric in Cao et al. [9] has an error that results in the wrong estimation of retrieval performance of the learnt hash functions. This error has propagated into many of the follow-up papers rendering the MAP numbers presented in them non-transferable. Most of the papers after HashNet have continued to use the same metric resulting in this situation. The code for the metric is provided as part of the open-sourced codebase of HashNet[1].

Everything until the computation of Precision@k ($k \in [1000]$) is correct. However, the reported metric computes AP@1000 as follows:

$$\text{AP@1000} = \frac{\sum_k P@k * rel(k)}{\sum_k rel(k)}$$

where $rel(k)$ is an indicator function if the sample at $k$-th position is relevant. $\sum_k rel(k)$ is the total number of relevant samples in all the 1000 retrieved samples. It should be noted that every query has around 1300 relevant documents and all of them can not be retrieved within 1000. MAP@k is just the mean of all the AP@k for all the queries. We explain the error using a simple example. Suppose a retrieval problem allows to retrieve 5 samples from a database where each query has 10 relevant samples. And for a given query, let us say method 1 gives retrieves the top-5 samples with the following relevance $[\mathbf{1}, \mathbf{0}, \mathbf{0}, \mathbf{0}, \mathbf{0}]$. And method 2 retrieved with samples with the relevance $[\mathbf{1}, \mathbf{0}, \mathbf{0}, \mathbf{1}, \mathbf{1}]$.

---

[1]https://github.com/thuml/HashNet/blob/master/caffe/models/predict/imagenet/predict_parallel.py#L20

Table 5: Image retrieval performance on ImageNet-100 using AlexNet. RMAP: Reported MAP@1000 as in HashNet [9]. CMAP: Corrected MAP@1000. Please see Appendix B for the details. RMAP and CMAP are not well correlated making the reported numbers from literature hard to compare.

| Method | 10 bits | | 16 bits | | 32 bits | | 48 bits | | 64 bits | |
|---|---|---|---|---|---|---|---|---|---|---|
| | RMAP | CMAP | RMAP | CMAP | RMAP | CMAP | RMAP | CMAP | RMAP | CMAP |
| HashNet [9] | 0.3721 | 0.1995 | 0.4634 | 0.2815 | 0.5915 | 0.4300 | 0.6548 | 0.5270 | 0.6542 | 0.5124 |
| LLC (Ours) | **0.4815** | **0.3086** | **0.5617** | **0.4305** | **0.6587** | **0.5565** | **0.6750** | **0.5749** | **0.6932** | **0.6000** |

Using the reported metric, method 1 has an AP@5 of 1.0 (with just 1 relevant sample out of 5). But method 2 with 2 more relevant samples along with the top retrieved sample and has an AP@5 of 0.7. Even with an objectively better retrieval, method 2 has a lower AP@5 score than method 1. This results in an unfair evaluation of methods where methods with poor recall but a few precise retrievals will outperform methods that have both high precision and recall.

The fix is simple and is just the use of standard MAP@k metric [23].

$$\text{AP@1000} = \frac{\sum_k P@k * rel(k)}{\min(\text{total relevant samples}, 1000)}.$$

The reason we need the $\min(\text{total relevant samples}, 1000)$ term in the denominator is that, with 1000 retrievals, it is impossible to retrieve all the 1300 relevant documents, so we divide by the most possible instead of all the relevant documents. With this metric, method 1 now has an AP@5 of 0.2, and method 2 has an AP@ 5 of 0.42 which reflects the reality of the retrieval. The only change needed is the replacement of "relevant_num" in this line[4] with "R" for the ImageNet-100 setting.

Table 5 presents a comparison of both MAP@1000 computed as reported, RMAP and corrected, CMAP for LLC and HashNet. This shows a stark drop in the MAP values along with a non-intuitive correlation between RMAP and CMAP making off-the-shelf comparisons harder. Lastly, the MAP computation from HashNet also skipped the AP values when none of the relevant documents were retrieved (which is extremely rare at 1000 retrieval samples) but the AP score should have been 0 for that particular case. We fixed that in both RMAP and CMAP metrics reported here.

## C   Hyperparameters

**Classification**    The first phase of the classification pipeline, ie., codebook learning, uses the hyper-parameters - SGD+Momentum optimizer, batch size of 256, cosine learning rate routine, and 100 epochs - used in training the standard 1000-way linear classifier using a ResNet50 [60, 35]. However, given the warm-starting of the second phase using the learnt backbone and the availability of the learnt codebook, the training routine of the second phase runs for about 25 epochs with a reduced learning rate of 0.01.

**Retrieval**    As we start with pre-trained models for retrieval. We use the same hyper-parameters as the second phase of the classification pipeline except for the learning rate. We use a much learning rate of 0.01 for AlexNet and 0.1 for ResNet50. The second phase of learning instance codes uses a reduced learning rate of 0.005 and 0.01 for AlexNet and ResNet50 respectively. The rest of the hyperparameters and training mechanisms are the same as the standard ResNet50 training.

---

[2]http://sdsawtelle.github.io/blog/output/mean-average-precision-MAP-for-recommender-systems.html#Average-Precision

[3]https://en.wikipedia.org/wiki/Evaluation_measures_(information_retrieval)#Average_precision

[4]https://github.com/thuml/HashNet/blob/master/caffe/models/predict/imagenet/predict_parallel.py#L42

# D Quantitative Evaluation of Hierarchy & More Visualizations

Quantitative evaluation of the discovered hierarchies is a hard problem because we can not directly compare ours to the original ImageNet hierarchy which is not binary unlike ours. However, one proxy way is to use the pair-wise inner-product heat map (see Figure 2) of 2048 dimensional representation as the base and compare using row-wise (class-wise) ranking metrics like Spearman's rank correlation coefficient in which the learnt 20-bit codebook has a mean coefficient across all class of 0.3 compared to 0.005 of a random codebook. Lastly, the number of unique class codes (ideally should be equal to $L$) learnt as part of the codebook plays an important role as the code collapse leads to loss of information about multiple classes resulting in unclear decoding when required.

It would be exciting to compare the discovered taxonomy to the original WordNet [41] hierarchy based on which the ImageNet-1K was curated. However, the major roadblock comes when we realize that our discovered hierarchy is binary while WordNet is $k$-ary making fair comparison almost impossible. We also explored the idea of cost-sensitive metrics [18] based on hierarchy to evaluate the classification but fell short due to the same limitation of unfairness in comparing binary to $k$-ary.

Lastly, we also wanted to see if the learnt bit codes/hyperplanes result in splitting the images by discovering (potentially interpretable) attributes as with previous works [47]. However, unlike previous works, which use simple features for encoding images, we learn a highly non-linear

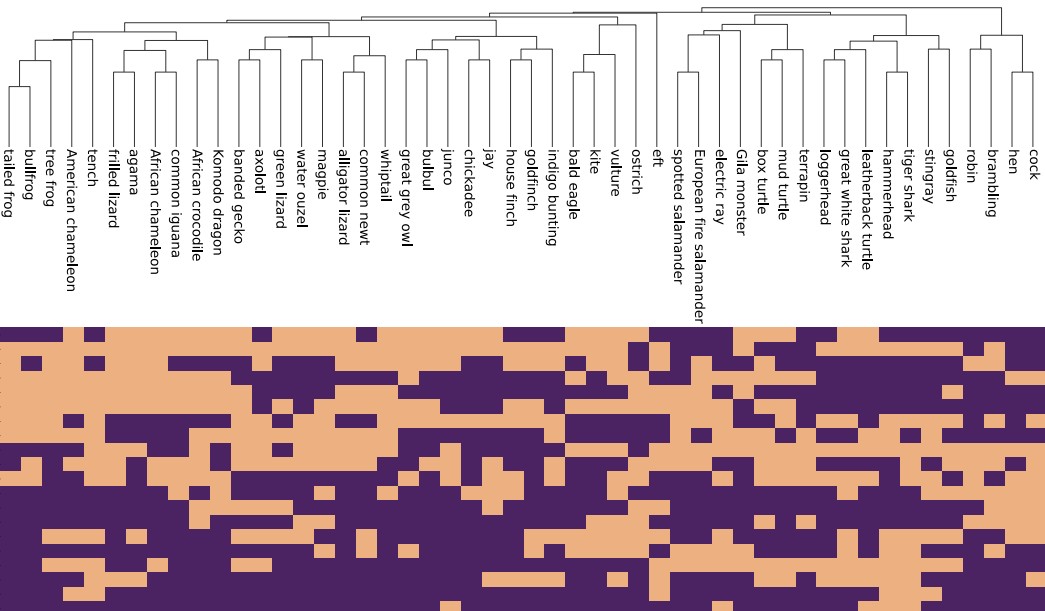

Figure 3: Discovered hierarchy on 50 classes of ImageNet-1K along with the corresponding class codes. Purple is $+1$ and beige is $-1$ in each class code corresponding to the species in the dendrogram.

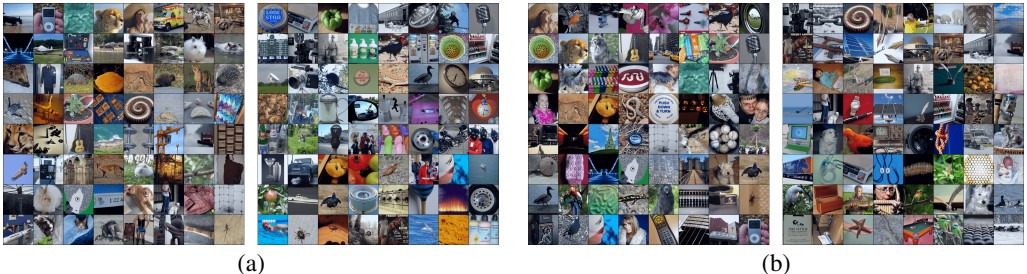

(a)                                                            (b)

Figure 4: The hyperplanes of bit numbers (a) 0 & (b) 16 visualized using the images being split on either side $+1, -1$ (left and right of the white bar in each of (a) and (b)), and are sorted using the probability. These splits do not show any trivial attributes being discovered, but often we find bits which do fine grained classification between close classes.

representation using deep neural networks. This resulted in learnt bit code hyperplanes that often split the images using highly non-linear, non-trivial, and non-interpretable attributes. Figure 4 shows the images on either side of the hyperplane of the corresponding bit sorted by the confidence of being $+1$ or $-1$ (probability from the logistic function for binary classification). While, with deeper analysis and further visualization, we might be able to deduce what is being learnt, but at the surface level, without explicit learning with priors, the discovered attributes are non-interpretable affecting zero-shot and few-shot capabilities of LLC based models.

# E    Out-of-Distribution (OOD) Detection Experiments

OOD detection for a multi-class classification model can be achieved by simple baselines that set a threshold on a heuristic based on the prediction probabilities [26]. For the classification models trained with ResNet50 on ImageNet-1K, we evaluate on two datasets which we consider as out-of-distribution to ImageNet-1K that has 50K samples as the in-distribution validation set. Our method uses a 20-bit classification model while the baselines use the pre-trained ResNet50 with linear classifier.

ImageNet-750 [55] is a long-tailed dataset of 750 classes with about 69K samples. These 750 classes were sampled from ImageNet-22K [17] ensuring no clash with the ImageNet-1K subset. All the instances in this dataset are OOD to a model trained on ImageNet-1K. MIT Places [65] dataset is aimed at scene recognition rather than object recognition, unlike ImageNet-1K. All the $\sim$18K samples in the validation set of Places365 were also treated as OOD to ImageNet-1K during this evaluation.

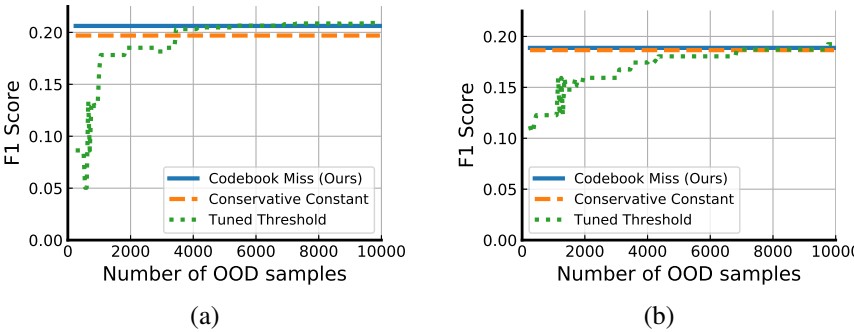

(a)                                                                      (b)

Figure 5: OOD detection performance when the validation set is not representative of the deployment setting for a) ImageNet-750 & b) MIT Places datasets evealuted on a ResNet50 model trained for ImageNet-1K.

As mentioned in Section 4.3, LLC based OOD detection looks at the encoded instance code and classifies a sample as OOD in case of a missing exact match in the learnt codebook. This does not require any OOD samples whatsoever apriori to the deployment. We consider two baselines: a) tuning of the threshold on the probability of maximum logit to maximize F1 score using a validation set with OOD examples [26], and b) setting a conservative threshold on the probability of maximum logit which is 1 standard deviation greater than the mean probability of the maximum logit using just 50 OOD examples. Note that the performance of the first baseline varies with i) the ratio of in distribution to OOD samples in the validation set that is being used for tuning the threshold, ii) the setup with a different validation to test out-of-distribution ratios.

Our primary focus is on the setting where the validation set used for tuning thresholds is not representative of the test set. In this setting the in distribution samples are constant, all the 50K samples. We only vary the number of OOD samples while tuning the threshold. The final OOD performance using the F1 score was measured on a test set with all the 50K in distribution samples along with a random 10K samples from the OOD set (ImageNet-750 & MIT Places).

Figure 5 captures the effectiveness of LLC based OOD detection in the setting where validation is not representative of test. Our out-of-the-box OOD detection method is at least as effective as the baselines. Remarkably, the tuning of the threshold to maximize F1 using a validation set requires at least 3000 OOD samples to even get close to our method which requires *no samples*.

The second setting is where the validation and test sets have the same number of OOD samples for a fixed in-distribution set. Here the validation and test phases have same number of re-sampled OOD instances. All the 50K in distribution samples are used for both validation and testing.

Figure 6 shows that LLC based OOD detection is as competitive as the tuned threshold baseline [26] which is optimizing F1 for the exact same ratios. These experiments help in arguing that LLC based classification models have a strong, sample-efficient inherent OOD detection capability. which can be used for sequential learning, to detect and continuously add new classes, along with robustness.

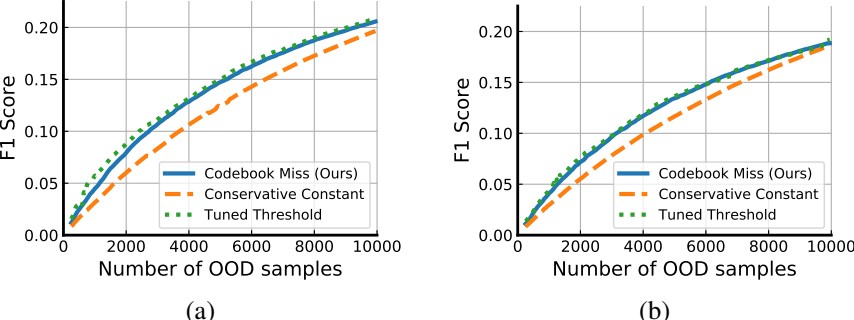

(a)                  (b)

Figure 6: OOD detection performance when the validation set is representative of the deployment setting for a) ImageNet-750 & b) MIT Places datasets evaluated on a ResNet50 model trained for ImageNet-1K.