# OpenReview forum: "LLC: Accurate, Multi-purpose Learnt Low-dimensional Binary Codes"
_NeurIPS.cc/2021/Conference — NeurIPS 2021 Poster_

### Official Review · Reviewer_z4qi · 2021-07-14

**Rating:** 4
**Confidence:** 4

**Summary:**

This work propose a method for learning low-dimensional binary codes for instance and classes. It is appealing in classification with large scale number of classes. The method is claimed super-efficient in learning and able to ensure nearly optimal classification accuracy. The learnt class code discovers some intuitive taxonomy over subset of classes selected from ImageNet. The method is applicable in image retrieval and out-of-distribution detection.

**Limitations And Societal Impact:**

Yes.

**Main Review:**

The paper need to be organized more seriously. The intuition for designing the loss in eq(2) is not well explained, why chose(P.F(x,theta)) over B(P.F(x,theta))? Is this kind of loss firstly proposed here, if not, then is it following existing works? What is the reason to choose this kind of loss?

There are too many words and too less figures. Some figures/ flow-charts are suggested to clearly show how to use the class code and the instance code in classification, image retrieval, and out-of-distribution detection.

Figure 1 shows the discovered taxonomy using the learnt class code, but there is no illustration on the knowledge discovered by the learnt instance code, figures are suggested to show the relationship between the class code and instance code.

On the experiments part. Most of the compared methods are before 2018, is there any other recently developed methods? Is the proposed method comparable with SOTA methods?

**Time Spent Reviewing:**

four

---

> ### Author Response · Authors · 2021-08-06
> **Rebuttal addressing all the concerns. Strong request to re-read the paper.**
>
> We thank the reviewer for their time and would like to address all the concerns raised through this rebuttal. We strongly think that the reviewer has not provided us with valid concerns for the given score. We urge the reviewer to revisit the paper and discuss any concerns with us.
>
>
> 1) **“Serious” paper organization**: We are sorry that you felt this way, we would be happy to incorporate any suggestions you might have about the paper organization as it contrasts with the other reviewers’ opinions. This would help us in further improving the presentation of the paper and cater to a wider audience.
>
> 2) **Intuition for Eq 2)**: It is a valid concern to have regarding the use of P.F(x,$\theta$) instead of g(x) = B(P.F(x, $\theta$) in Eq 2) as g(x) is essentially what we need for the instance codes. The main reason for the 2-phase setup and not directly going for g(x) is the *hardness of learning both binarized class and instance codes* in one optimization step.
>
> - Optimization in binary space is already a *hard discrete* problem (as seen by many other works like binary neural networks etc.,) and when working with such low-dimensional spaces, SGD needs overparameterization and relaxation to learn and have stable training. When we experimented with g(x) instead of P.F(x,$\theta$) in Eq 2) the algorithm failed to learn anything useful and the loss has stagnated. This led us to decouple the learning of class codes and then use the learnt class codes to further learn the instance codes using Eq 3).
>
> - This is a *novel loss in the context of binary code learning*. The previous works have tried to directly learn g(x) but through an unbinarized C, which has similar reasoning to our design choices. However, these previous methods do not yield class codes nor do all the instance codes map to their corresponding class codes, breaking the needs of both efficient classification and OOD.
>
> - Overall, the proposed 2-phase framework is a well-experimented design to support stable training while improving accuracy.
>
> 3) **Figures/Flow-charts**: Thank you for the suggestion. We are happy to incorporate a teaser figure with an intuitive explanation of the LLC algorithm and its use in downstream applications. We already have presented *clear descriptions* (as R1 (vExU) points out as well) in each subsection of the main paper along with algorithms in Appendix A. We would also like to point out more visualizations in the appendix. Lastly, there is a space constraint and this will be fixed with the additional page after acceptance.
>
> 4) **Knowledge discovered by instance codes**: Illustrating knowledge is hard. In LLC, the instance codes are learnt through class codes as mentioned multiple times (The 2-phase LLC pipeline). The optimization problem helps distill all of the “knowledge” from the class codes to the instance codes. This transfer can be argued with the help of Exact Decoding (ED) based classification experiments in Table 1. ED expects the instance to have the exact binary code as the ground truth class, results where we get ~69% (using 20 bits) compared to an all-powerful baseline of ~77% which uses 2048 dimensional real numbers and an exhaustive maximum inner product search.
> In short, the knowledge discovered by instance codes is similar to the class codes as instance codes are designed to be learnt via the learnt class codes.
>
> 5) **Comparable to SOTA**: As pointed out by R1 (vExU), we did a clear and extensive literature review on binary class and instance codes. This also includes the relevant SOTA methods. We request the reviewer to read the literature review and provide us with any relevant SOTA methods we might have missed.
> - In our survey, we found that the classification aspects of this paper were only explored in the past and we covered them exhaustively.
> - Retrieval is still an active area via deep supervised hashing (DSH). HashNet (ICCV 2017) is seminal in the field of DSH and most techniques are better-engineered versions of it. GreedyHash (NeurIPS 2018) is one such technique which as shown in Table 3 significantly improves over HashNet and remains a competitive benchmark even now. GreedyHash and HashNet only differ in the way they optimize for the instance codes and the underlying techniques of GreedyHash are still being used in DSH. GreedyHash and HashNet had reproducible codes for ImageNet-100. Lastly, efficient retrieval was a byproduct of our LLC pipeline which we utilized to showcase the breadth of possibilities with learnt class and instance codes.
> - We believe this is the first setup to perform OOD with binary codes and the baselines we use are still the most competitive ones across the board for OOD detection.
>
> We earnestly request the reviewer to re-read the paper, rebuttal and have a discussion with us for any further clarification. We hope this helps in increasing the score to facilitate acceptance.

---

### Official Review · Reviewer_BkS5 · 2021-07-15

**Rating:** 6
**Confidence:** 4

**Summary:**

This work proposes a two-phase method for learning low-dimensional binary codes via the standard classification task without side-information. Specifically, binary codes for classes are learned via a surrogate classification task in the first phase. On top of that, instance codes are learned using the Error-Correcting Output Codes approach.

**Ethics Review Area:**

["I don’t know"]

**Limitations And Societal Impact:**

According to question 1(b) in the Checklist, the authors describe the limitations and give potential solutions.

**Main Review:**

-      Pros
* Well written paper with extensive experimental results.
* Improvements are found on three downstream tasks in terms of computational efficiency.
-      Cons:
* According to Equ. (2) and (3), both class codes and inference codes are learnt by minimizing the classification-related loss. However, there are some other important criteria of binary codes, such as low-bit correlation, robustness against transformations, and neighboring relationships. The authors should explain how those criteria are ensured in the proposed work?
* According to Equ. (2), the codebook B(C) is learned by minimizing the multiclass prediction scores and the ground-truth one in Phase 1. However, since B(C) should be the only variable in Equ. (2), the authors should explain the initialization of g(x). And how does it ensure the training stability, especially in the first several epochs, when the instance embeddings are not discriminative?
*Some typo errors could be found in the paper. For example, in Line 257, the word “iff” should be corrected as “if”.

**Time Spent Reviewing:**

4

---

> ### Author Response · Authors · 2021-08-06
> **Rebuttal addressing all the 3 concerns. Request to revisit the paper.**
>
> We thank the reviewer for appreciating the writing and experimental evaluation of the paper along with the gains it provides for the downstream tasks. In this rebuttal, we would like to address all the 3 concerns raised. We are happy to have a discussion to facilitate further clarifications. Also, we believe that the reviewer has not provided any strong reasons/concerns + actionable feedback for the rating provided and hope that this rebuttal helps in clearing things up.
>
> 1) **Incorporating other criteria of binary codes**:  We strongly disagree with the reviewer on this point. As mentioned multiple times in the paper, we explicitly made sure not to use any of the traditional criteria of building the binary codes as mentioned in the past work. We aim to provide low-dimensional binary codes that can help in important tasks like classification/retrieval. We wanted to see what all could be learnt without using any side information (like attributes) and priors for constructing the binary codes (like the ones reviewer mentioned). These are well-established tasks and binary codes have been used in these contexts (and we compare against them). **So, why should “every” binary coding method satisfy the other properties?** We agree that in some settings, the other criteria might be important, but we choose to focus on the performance in important settings that are well established in the area.
>
>
> Even though we do not use any criteria for binary codes, they naturally evolve from our learning algorithm.
> - We learn the binary codes through a classification loss/task, with SGD and cross-entropy loss, we expect and observe that the classifiers are distinct and well separated to maximize accuracy. This satisfies the low-bit correlation aspect for most of the classes. Also, to ensure high accuracy with such low dimensions the learnt embeddings *shouldn’t be wasting bits*.
>
>
> - Neighbouring relationships essentially mean the hierarchy. As shown by the discovered taxonomy (Fig 1) and experiments on classification, retrieval, and OOD (Section 4), the binary codes are *well separated and intrinsically learn the hierarchy* (eg., embeddings of tree frog and bullfrog naturally are much closer to each other than say embedding of terrapin). The key reason for this is that neural networks are continuous function approximators and tend to keep semantically similar things closer leading to the discovery of intrinsic hierarchy as claimed.
>
> - We are not clear about the robustness against the transformations part and would request clarification from the reviewer. If they mean the transformation of the input, then we would like to point out that our training routine has the standard data-augmentation as part of it and we can add more input transformations as deemed necessary.
>
> - Finally, as mentioned in *Lines 392-393*, our framework can handle/incorporate (if needed) other criteria via additional regularization losses.
>
> But to emphasize again, not every binary code needs to satisfy the other properties, as long as they satisfy the task goals they set out for, and in this case, we observe strong performance on three downstream tasks.  We hope that the paper is judged on solving the problem we consider (i.e., if the problem is important enough and if the method advances state-of-the-art for that problem) instead of problems we don’t even intend to solve.
>
>
> 2) **Regarding g(x) in Eq 2)**: This is a misunderstanding at the reviewer’s end and is factually wrong. We would respectfully ask the reviewer to revisit Eq 2). *Eq 2) does not claim B(C) is the only learnable variable*, but rather minimizes loss for *all the parameters* involved ie., C, P, and \theta. So, we need not have to worry about the initialization of g(x) as everything is learnt end-to-end. This also helps us argue that the training is stable because the *instance embeddings are also being learnt simultaneously*. Also, we do not use g(x) in Eq 2) but rather use the non-binarized version of the same. Please note that after phase 1, we further fine-tune P, $\theta$ in phase 2 via Eq 3) which ends up giving us g(x). Lastly, during the experiments, *we never faced any training instability* in either of the phases of training, ie., at the end of phase 1 with 20-dim *real* instance embeddings and binarize class codebook we get maximum inner product search (MIPS) accuracy of 72.5% and after further fine-tuning with phase 2, now the MIPS accuracy (which is same as MHD in this case) increases to 74.5% but uses *20-bi*t instance and class binary codes
>
> 3) **“Iff” & typos**: We indeed meant *“iff”  (i.e., “if and only if”)* because the goal of deep supervised hashing (DSH) is to encode *only* instances from the *same* class to be closeby. After the submission, we fixed the few remaining typos which should reflect in the next version eg., “Retrival” in Table 3 caption.
>
> We believe that these responses can help in clarifying the doubts the reviewer had regarding our paper and hope that they increase the score and facilitate acceptance. We would be very happy to discuss any further clarifications to alleviate your concerns on the LLC paper.

---

### Official Review · Reviewer_vExU · 2021-07-17

**Rating:** 7
**Confidence:** 3

**Summary:**

Low-dimensional binary codes are important for a variety of large-scale ml tasks, especially retrieval. This paper proposed a method called LLC for learning semantially meaningful low-dimensional biary codes. Compare to the existing literature, LLC can learn both class and instance codes together without side-information:  (1) couple the learning process of both feature and class embedding and capture the sematic structure better (2) naturally benefits downstream tasks like OOD.

**Limitations And Societal Impact:**

Yes.

**Main Review:**

General Comments

The paper proposes a neat algorithm for low-dimensional binary code learning. Although it is a (very) well-studied area, authors present the related work or literature clearly and at the same time show the novelty of their proposal. I understand the paper has already done a wide range of applications and extensive ablations (breath), but I think there's a potential to improve the depth of the work -- what's special/principled about LLC's learnt binary codes, not the algorithm of how to learn it but the learnt codes? Details are in pros and cons.

Pros:

- Writing/Pitching: The problem is well-motivated. Literature review for learning binary class and instance codes is very clear.
- Algorithm: It is a smart 2-stage algorith, which first learns binary codes for class and instance together and then using the learnt class binary codes as the better than random "pseudo" labels to "correct" binary codes for instance.
- Utility: Evaluations on classification, retrieval and especially OOD provide good coverage of popular applications; ablations and visualizations are helpful.

Cons:

- Algorithm integrity: There are two stages for learning class and instance binary codes, the paper evaluates how much lift from stage 1 only to stage 1+stage 2. I'm curious how impactful is the quality of class binary codes learnt from stage1 to instance binary codes learnt at stage 2? How to fix the problem if stage1 learns suboptimal/bad representations in some other applications or datasets?
- Depth: Authors have analyzed class binary codes qualitatively and find they learnt semantic structure because there's hierarchy for hash codes. That's probably the reason why stage 2 is helpful. But I think it would be very interesting to dig further on what's special about instance binary codes? Are they more uniform, more compact? Why they are better than the ones the baseline generated on retrieval/classifcaiton and especially ood?


**Time Spent Reviewing:**

3

---

> ### Author Response · Authors · 2021-08-05
> **Thank you for the helpful review. Requested Clarifications**
>
> We thank the reviewer for the positive review and constructive feedback. We appreciate the reviewer for understanding the core idea of the paper and its potential in a wide array of applications. We would be grateful if the reviewer can help other reviewers realize the main takeaways thus supporting the rebuttal and the paper’s acceptance.
>
> We try to clarify the questions here:
> 1) **Specialty of LLC’s learnt codes**: LLC’s learnt codes are special because they are able to learn the *intrinsic semantic information* which further helps in discovering an intuitive hierarchy. This is all done without using any auxiliary information. The key aspect is that we are able to learn unique class codes in such a low-dimensional space while capturing the semantics. Given the semantic richness, coupled with the tightness, LLC's learnt codes provide a path towards near sub-linear classification, retrieval and OOD detection among others.
>
> 2) **Impact of class codes on instance codes**: This is the centerpiece of our classification experiments shown in Tables 1 and 2. The quality of the class codes can be measured in two ways: **a)** Number of unique codes learnt & **b)** The intrinsic semantic information captured (measured either via hierarchy or some proxy accuracy measures). We show that both these factors affect instance code learning which further impacts classification accuracy.
>
> 3) **Fixing the problem of sub-optimal codes from stage 1**: Before we answer this question, we would like a clarification as to what the reviewer means by suboptimal/bad representation. In this response, we assume it means being bad in one of the two measures mentioned above (a & b). In our experiments, we noticed that increasing the length of the binary code usually helped in fixing the issues as that also helps in better optimization. We also show that we can start with a worse codebook like SVD/CCA (Line 333-338) and use stage 1 of LLC to eventually get to a good codebook. However, we acknowledge that there might be datasets with no inherent hierarchy making the codebook learning hard, but we think in that case there is potentially nothing optimal and the only measure is to have as many distinct codes as possible.
>
> 4) **About instance binary codes**: As the reviewer has mentioned, phase 2 basically relies on phase 1 ie, instance codes are learnt via the learnt class codes. Whatever makes the learnt class codes special translates to instances codes as the formulation expects the instances of a given class to have an exact (or nearly exact) same binary code as its corresponding class. This helps them be more compact (as much as the class codes) while also having a “cube” (most instance codes are within a small hamming distance) of influence around the class code itself. They are better than the baselines in classification because the quality of the baseline class codes is poorer while retrieval baselines do not force instances to map to a class code exactly like we do but only enforce close encodings when feasible. Lastly, because of the tight and exact mapping to the class codes, it makes sense to use the unassigned codes (as most instance codes of every class are mapped to class codes) to potentially detect novel classes. However, the level of strictness can also be varied based on how noisy the ground truth labeling of the dataset is thus providing a handle on the trade-off between in vs out of distribution detection.
>
> We hope that this rebuttal further solidifies your positive outlook on the LLC paper and we are happy to discuss if you need any further clarifications to facilitate the acceptance of the paper.

---

### Official Review · Reviewer_Kaxs · 2021-08-12

**Rating:** 7
**Confidence:** 2

**Summary:**

This paper studies the problem of learning binary vector representations of instances and classes. We pick a small dimension k (the paper uses k=20) and aim to learn a k-dimensional binary vector for each input and each label. Compressing the data into small-dimensional binary codes like this is challenging, but provides many benefits in terms of the space usage (the representations are very small) and the time to do lookups and find nearby neighbors.

The paper presents a new method called LLC which simultaneously finds codewords for both inputs and labels without using any side-information. They claim (and I have no reason to doubt this) that this is the first time a method for this has been designed. LLC consists of two phases:

In the first phase, the LLC method learns a codebook (low-dimensional bianry code for each label) by using the popular Straight-Through Estimator technique to do empirical risk minimization.

In the second phase, the ECOC framework is used to classify the inputs according to this codebook. The algorithm treats each of the k bits of the code completely separately, and hence solves k disjoint binary classification problems instead of one multi-class (with L or 2^k classes) problem. This gives an additive 2% improvement to accuracy over just solving the optimization problem to classify all k bits together, since out of the box optimization finds fairly suboptimal solutions when trying to classify a length k vector all at once. (In other words, the second phase uses a heuristic where sepaartely learning each of the k bits works better than learning all k at once for the current optimization software which is well-suited to doing binary classification.)

The resulting classification accuracy seems quite good: 68.82% of inputs are assigned exactly the same binary code as their label, and 74.57% are assigned a binary code which is closer to the code of their label than any other label.

The result of this binary code classification can now be used in a number of applications, including:
- the binary codebook can be naturally used to make a taxonomy visualization for the labels.
- the system can be used for efficient (in terms of representation size and running time of retrieval) multi-class classification.
- out-of-distribution detection can be done by seeing whether an input is mapped to the same codeword as any label. (I'm confused about how effective this can be when the original inputs are only correctly mapped to the codeword of their label 69% of the time.)

**Limitations And Societal Impact:**

Not really discussed, though I'm not sure what there is to say about this.

**Main Review:**

The paper is generally well-written, although it refers to a number of different techniques as "standard" instead of explaining them in more detail, and I wasn't familiar with some of them, which made it hard to understand some steps of the algoritm. There are some English grammatical issues (including many sentences with extra commas added, and many sentence fragments in more technical sections), and I recommend that the paper is carefully reread before it is published.

Overall, I think this paper is solving an interesting problem, and it seems to have good accuracy. The main new ideas seem to be (1) using a somewhat clever heuristic to pick the codebook instead of using random codewords, and (2) splitting the multi-class classification task into k disjoint binary classification tasks. Otherwise, standard tools from the area are used. I think these are fairly natural ideas for this problem (and not huge breakthroughs or anything), but they nonetheless perform quite well. It's surprising to me that such an approach hasn't been taken before.

I have some questions for the authors:

1. In Section 4.1.1, it says "we evaluate the hamming 205 distance with all the L class codes and output the class with the least hamming distance". What happens if there is a tie, and multiple class codes have the same Hamming distance? Since we are dealing with only 20-dimensional vectors, I imagine this must be common (is it?). (Also note: Hamming should be capitalized since it's someone's name.)

2. There isn't much discussion of the running times in practice/applications. How long did your code run for? How much time is saved by making phase 2 take time independent of L?

3. Phase 2 seems to only consider two extremes: classify all k bits at once, or just one at a time (which does better). What if an intermediate option were used (such as classifying pairs of bits at the same time, i.e., transform the problem into k/2 independent 4-class classification problems)?

4. One could imagine doing Phase 2 instead by just using a high accuracy classifier which totally ignores the codebook (and then afterwards looking up the codeword from the codebook for each label). How much more accurate would this be? How much slower would it be? Are there other downsides of this approach?

**Time Spent Reviewing:**

2

---

> ### Author Response · Authors · 2021-08-14
> **Thank you for the thorough review. Requested Clarifications.**
>
> We thank the reviewer for the positive review and thorough feedback. We appreciate the reviewer for understanding the core idea of the paper and its potential in a wide array of applications. We try to clarify your concerns here:
>
>
> 1) **Writing**: Thanks for the comments that further help in the presentation of the paper. We will go through the paper again and fix the issues mentioned in the camera-ready version of the manuscript. It would also be helpful if you could point out the particular instances of “standard” which were unclear.
>
> 2) **Effectiveness of the proposed OOD detection**: This is a valid concern. However, we are measuring the metrics like F1 which capture both the *in distribution* (IND) and *out of distribution* performance simultaneously. The effectiveness of our proposed method comes from strong IND as well as OOD detection performance. Another key point is that our method uses *no* validation data to tune the threshold and is also agnostic to the variation of IND to OOD ratios in validation and test sets.
>
> 3) **Tiebreak for Minimum Hamming Distance (MHD)**: This is another valid concern while using MHD. In case of a tie in Hamming distance, we randomly pick one of the classes in the current setup. However, we can also make use of the raw probabilities assigned to each bit for breaking ties in these cases. MHD gets to 74.57% accuracy compared to the all-powerful baseline with 77%. This shows that the clashes are not that frequent but given the existence of probabilities, we can make finer tiebreaking choices. Thanks for pointing out the mistake with “hamming”, we will fix it in the next version.
>
> 4) **Discussion about run times**: Thanks for this question. With a large-scale image dataset like ImageNet-1K, most of the computational cost of the model is owed to the ResNet50 backbone, both during training and inference. As mentioned in Line 251, the gains in such a scenario will only be profound and visible in wall clock times at the scale of millions of classes. As mentioned in future work (line 401-403), we expect this to be very useful in extreme classification settings where the featurizers are not the most expensive parts. Due to the same reasons, the gains in phase 2 due to sub-linear costs are not quite visible for ImageNet-1K. We strongly believe that this is a promising future research direction to showcase the efficacy of LLC in terms of computations gains as well as accuracy trade-offs.
> For further details on default training costs (in epochs) for both phases, please see Appendix C.
>
> 5) **About extreme decoding schemes**: We discuss this in Lines 382-384. Designing intermediate decoding schemes is of high value and is a logical next step. We are actively trying to come up with something as accurate as MHD but with sub-linear costs like ED. This also involves modeling decoding as a sequential learning problem to make use of hierarchy during decoding. We are happy to discuss further this aspect because of its high impact and importance.
>
> 6) **High accuracy classifier for Phase 2**: We are not completely sure about what the reviewer means by this. Is the question about using a strong non-linear classifier on top of real-embeddings or binary codes of instances? We are also not sure about what the reviewer means by “afterwards looking up the codeword from the codebook for each label”. We are not clear on how this helps efficiency in general. The two efficiency gains LLC brings are due to a) use of binary codes which are an order of magnitude faster than real numbers and b) due to the sub-linear dependence on the number of labels. Both these are vital for extreme efficiency, so the use of a strong non-linear classifier (with real numbers) will increase the decoding costs at a marginal increase in accuracy as we show that MHD is only about 2.5% less accurate than the real-valued baseline.
>
> Along with addressing your concerns, we would also like to point out that we do discuss the limitations of LLC in Section 5. Lastly, we are not using a clever heuristic to pick a codebook, but rather are learning it from scratch with no side information whatsoever. We hope that this rebuttal further solidifies your positive outlook on the LLC paper and we are happy to discuss if you need any further clarifications to facilitate the acceptance of the paper.

---

> > ### Comment · Reviewer_Kaxs · 2021-08-30
> > **Thanks for the response**
> >
> > Thanks for the response! That all makes sense to me. I think reviewer 5Yju does a good job of expressing an important point that I didn't articulate well: it's suprising here that the "all-powerful baseline" only achieves 77% (and maybe indicates more work should go there), but in light of this, the fact that you can achieve almost the same accuracy is impressive.

---

> > > ### Author Response · Authors · 2021-08-31
> > > **Response regarding all-powerful baselines**
> > >
> > > We would like to clarify what we mean by all-powerful baseline. An all-powerful baseline in this context has the same backbone (ResNet50) and uses a full-blown real-valued linear classifier that operates on a 2048 dimensional real vector to generate probabilities for 1000 classes. The 77% is the accuracy of a baseline ResNet50 model on ImageNet from most of the well-used libraries (Pytorch, TF etc.,). The baseline numbers change with the use of different architectures like MobileNets, DenseNets, Transformers etc.,
> > >
> > > We are only comparing with a 77% ResNet50 baseline because all of our experiments use a ResNet50 backbone for fair comparison. Hope this clarifies any doubts you have regarding the baseline.

---

### Official Review · Reviewer_5Yju · 2021-08-29

**Rating:** 7
**Confidence:** 2

**Summary:**

This paper gives a new empirical approach for learning schemes for compressing high-dimensional neural representations into binary codes over a small number of dimensions. Given labeled examples and a fixed neural architecture with d-dimensional output, they give a two-phase algorithm for learning weights for the network, a low-dimensional projector to a k-dimensional subspace of R^d, and a binary "codebook" matrix B(C) mapping the k-dimensional subspace to R^L, where L is the number of classes. The goal is to do this for k as close to log_2(L) as possible while mostly preserving test accuracy.

Concretely, the embedding g maps any input x to the Boolean vector given by the entrywise sign of the k-dimensional projection of the network's output under x, and B(C) maps any such g(x) to an L-dimensional vector corresponding to the scores of the different classes. In the first phase of the algorithm, they simply run ERM to simultaneously find some weights, projector, and codebook, using a standard heuristic to get around the fact that B(C) is discrete-valued. In the second phase, they train the weights and projector further by essentially encouraging the embedding to align with the codebook.

The distinguishing feature of this approach is that the codebook and embedding are learning in conjunction rather than separately. Empirically over ImageNet-1K, they found that under two natural decoding schemes, the resulting classifiers beat a number of baseline approaches that separately constructed a codebook and then learned an embedding. Additionally, the accuracy is only ~3% less than the standard ResNet50 baseline.

As other applications, they showed that for image retrieval, the learnt embedding can be used as a hash function, outperforming HashNet and GreedyHash. They also gave an application to out-of-distribution detection which is comparable to certain baselines but notably does not need samples from the out-of-distribution domain.

**Limitations And Societal Impact:**

Yes, the authors adequately address the limitations, and as they note in the checklist, their work inherits the upsides and downsides of other efficient ML research.

**Main Review:**

The approach put forth in the submission seems quite effective, given that they are able to compress down to 20-bit representations while taking a hit of only 3% relative to the compression-free baseline of 77% for ImageNet-1K. It would be interesting to see how this sort of drop in performance manifests when there are even more classes, some of them potentially underrepresented, e.g. ImageNet-22K as mentioned on the last page.

That said, it seems a bit weird that just running ERM on the weights/projector/codebook already beats the other codebook baselines to which the authors were comparing, and I'm slightly worried that this is simply because the baselines weren't very compelling to begin with. If there aren't better baselines out there, then should I think of the particular instance and class code parametrization in (1) as already giving a nontrivial win over previous works? It does feel like (1) is a very natural thing to try.

Nevertheless, the applications to retrieval and OOD detection, as well as the discovered taxonomy findings, suggest that the usefulness of LLC extends beyond just classification, and for these reasons I would recommend acceptance.

**Time Spent Reviewing:**

5

---

> ### Author Response · Authors · 2021-08-29
> **Thank you for the thorough review. Requested Clarification.**
>
> We thank the reviewer for the positive review and thorough feedback. We appreciate the reviewer for understanding the core idea of the paper and its potential in a wide array of applications. We try to clarify your concerns here:
>
> **Large-scale datasets (# classes):** We agree with you that extending LLC to a large number of classes - potentially millions as mentioned in future work - is a logical next step. Working with long-tailed data is a part of it as most million-scale web datasets are long-tailed as seen in extreme classification, real-world object recognition, etc., We would also like to point out that while MHD is an efficient scheme with only ~3% drop in accuracy, the trade-off ED scheme provides is vital for efficient classification at million-billion scale given its sub-linear costs (nearly independent of L). We hope that these gains and trade-offs work at scale and we expect to see this in immediate future work.
>
> **Baselines:** This is a valid concern. To the best of our knowledge, our literature review on codebook learning is exhaustive.  The prior literature on codebook construction often relied on hierarchy or other modalities which were shunned away for random codebooks given the extra cost involved vs. the accuracy trade-offs. The baselines we use include random codebook as well as two methods (CCA and SVD) that inherently encode the hierarchy.  We believe and assure you that these are compelling baselines.
>
> **Natural thing:** Yes, (1) is the natural thing to do because it is a general purpose formulation and not the solution. The problem formulation in (1) is the *general formulation for the problem setup* and each of the baselines also fits into it. We can solve (1) in multiple ways like presetting the class codes (random or otherwise), pairwise optimization (Line 370-374) and using smooth relaxations for binarization function (sigmoid/tanh) etc., all of which fail to have high accuracy or suffer from code collapse due to training instabilities. LLC is the first solution that learns both class codes (codebook) and instance codes stablely with no side information whatsoever. LLC's innovation comes in solving the problem using the techniques in its 2-phase routine. Current design choices of LLC came after thorough experimentation where we realized the earlier solutions are not good enough. *You should be thinking of LLC as a solution for the general formulation (1) that is giving a nontrivial win over previous works.*
>
> Thanks for recommending acceptance. We hope that this rebuttal further solidifies your positive outlook on the LLC paper and we are happy to discuss if you need any further clarifications to increase the score.

---

> > ### Comment · Reviewer_5Yju · 2021-09-02
> > **Thanks for the response!**
> >
> > Thanks for the detailed response, in particular for clarifying the innovative components of LLC and the absence of better baselines! Given that (1) is the general formulation for the problem, I do still think it's a bit surprising that ERM already beats other baselines, but it does seem like the potential impact of this work in handling datasets where L is much larger justifies a higher score, so I will upgrade to a 7.

---

> > > ### Author Response · Authors · 2021-09-02
> > > **Thank you**
> > >
> > > Dear reviewer,
> > >
> > > Thank you for engaging in the discussion and increasing the score. Thanks for your time and effort.
> > >
> > > Regarding ERM beating all the baselines: Yes, it is a simple solution but we found it to be super effective as the results suggest. We can further improve and build on these ideas while not relying on side information in the future for codebook and instance code learning.

---

### Decision · Program_Chairs · 2021-09-27

**Decision:**

Accept (Poster)

**Comment:**

The problem considered in this paper, namely, of learning compression schemes for high dimensional neural networks, is a very natural and important problem. The paper achieves new, state-of-the-art results with new techniques for this natural problem. While the proposed technique lacks theoretical grounding, the empirical results are quite impressive. In discussions the authors sufficiently clarified concerns the reviewers had about the lack of good baselines for this field. Overall I think this is an interesting paper on an interesting topic and worthy of inclusion.